# *muscat* detects subpopulation-specific state transitions from multi-sample multi-condition single-cell transcriptomics data

Helena L. Crowell [1,2], Charlotte Soneson[1,2,3,6], Pierre-Luc Germain[1,4,6], Daniela Calini[5], Ludovic Collin[5], Catarina Raposo[5], Dheeraj Malhotra[5] & Mark D. Robinson [1,2✉]

Single-cell RNA sequencing (scRNA-seq) has become an empowering technology to profile the transcriptomes of individual cells on a large scale. Early analyses of differential expression have aimed at identifying differences between subpopulations to identify subpopulation markers. More generally, such methods compare expression levels across sets of cells, thus leading to cross-condition analyses. Given the emergence of replicated multi-condition scRNA-seq datasets, an area of increasing focus is making sample-level inferences, termed here as differential state analysis; however, it is not clear which statistical framework best handles this situation. Here, we surveyed methods to perform cross-condition differential state analyses, including cell-level mixed models and methods based on aggregated pseudobulk data. To evaluate method performance, we developed a flexible simulation that mimics multi-sample scRNA-seq data. We analyzed scRNA-seq data from mouse cortex cells to uncover subpopulation-specific responses to lipopolysaccharide treatment, and provide robust tools for multi-condition analysis within the *muscat* R package.

[1] Department of Molecular Life Sciences, University of Zurich, Zurich, Switzerland. [2] SIB Swiss Institute of Bioinformatics, Zurich, Switzerland. [3] Friedrich Miescher Institute for Biomedical Research and SIB Swiss Institute of Bioinformatics, Basel, Switzerland. [4] D-HEST Institute for Neuroscience, Swiss Federal Institute of Technology, Zurich, Switzerland. [5] F. Hoffmann-La Roche Ltd., Pharma Research and Early Development, Neuroscience, Ophthalmology and Rare Diseases, Roche Innovation Center Basel, Basel, Switzerland. [6] These authors contributed equally: Charlotte Soneson, Pierre-Luc Germain. ✉email: mark.robinson@imls.uzh.ch

A fundamental task in the analysis of single-cell RNA-sequencing (scRNA-seq) data is the identification of systematic transcriptional changes using differential expression analysis[1]. Such analyses are a critical step toward a deeper understanding of molecular responses that occur in development, after a perturbation, or in disease states[2–5]. Most current scRNA-seq differential expression methods are designed to test one set of cells against another (or more generally, multiple sets together), and can be used to compare cell subpopulations (e.g., for identifying marker genes) or across conditions (cells from one condition versus another)[6]. In such statistical models, the cells are the experimental units and thus represent the population that inferences will extrapolate to.

Given the rise of multi-sample multi-group scRNA-seq datasets, where measurements are made on hundreds to thousands of cells per sample, the goal shifts to making sample-level inferences (i.e., experimental units are samples), in order to account for sample-to-sample as well as cell-to-cell variability and make conclusions that extrapolate to the samples rather than cells. We refer to this generally as differential state (DS) analysis, whereby a given subset of cells (termed hereafter as subpopulation) is followed across a set of samples (e.g., individuals) and experimental conditions (e.g., treatments), in order to identify subpopulation-specific responses, i.e., changes in cell state. DS analysis: (i) should be able to detect changes that only affect a single cell subpopulation, a subset of subpopulations or even a subset of cells within a single subpopulation; (ii) is intended to be an orthogonal analysis to clustering or cell subpopulation assignment; and, (iii) can be considered a separate analysis to the search for DA of subpopulations across conditions.

We intentionally use the term *subpopulation* to be more generic than cell *type*[7,8], which itself is meant to represent a discrete and stable molecular signature; however, the precise definition of cell type is widely debated[2,3]. In our framework, a subpopulation is simply a set of cells deemed to be similar enough to be considered as a group and where it is of interest to interrogate such sets of similarly defined cells across multiple samples and conditions. Therefore, cells from a scRNA-seq experiment are first organized into subpopulations, e.g., by integrating the multiple samples together[9] and clustering or applying a subpopulation-level assignment algorithm[10] or cell-level prediction[11]; clustering and manual annotation is also an option. Regardless of the model or the uncertainty in the subpopulation assignment, the discovery framework we describe provides a basis for biological interpretation and a path to discovering interesting expression patterns *within* subpopulations across samples. Even different subpopulation assignments of the same data could be readily interpretable. For example, T cells could be defined as a single (albeit diverse) cell subpopulation or could be divided into discrete subpopulations, if sufficient information to categorize the cells at this level of resolution is available. In either case, the framework presented here would focus on the subpopulation of interest and look for expression changes *across* conditions. This naturally introduces an interplay with the definition of cell types and states themselves (e.g., discrete states could be considered as types) and thus with the methods used to computationally or manually classify cells. Overall, our goal here is to explore the space of scRNA-seq datasets with several subpopulations and samples, in order to understand the fidelity of methods to discover cell state changes.

It is worth noting that extensive workflows for DS analysis of high-dimensional cytometry data have been established[12–15], along with a rich set of visualization tools and differential testing methods[13,16–18], and applied to, for example, unravel subpopulation-specific responses to immunotherapy[19]. Notably, aggregation-based methods (e.g., representing each sample as the median signal from all cells of a given subpopulation) compare favorably in (cytometry) DS analysis to methods that run on full cell-level data[17]; however, in the cytometry case, only a limited range of cell-level and aggregation approaches were tested, only simplistic regimes of differential expression were investigated (e.g., shifts in means), and the number of features measured with scRNA-seq is considerably higher (with typically fewer cells).

In scRNA-seq data, aggregating cell-level counts into sample-level "pseudobulk" counts for differential expression is not new; pseudobulk analysis has been applied to discover cell-type-specific responses of lupus patients to IFN-β stimulation[20] and in mitigating plate effects by summing read counts in each plate[21]. In these cases, pseudobulk counts were used as input to bulk RNA-seq differential engines, such as *edgeR*[22], *DESeq2*[23], or *limma-voom*[24,25]. Also, non-aggregation methods have been proposed, e.g., mixed models (MMs) were previously used on cell-level scRNA-seq expression data[26] to separate sample and batch effects, and variations on such a MM could be readily applied for the sample-level inferences that are considered here. Various recent related developments have taken place: a compositional model was proposed to integrate cell type information into differential analysis, although replication was not considered[27]; a multivariate mixed-effects model was proposed to extend univariate testing regimes[28]; and, a tool called *PopAlign* was introduced to estimate low-dimensional mixtures and look for state shifts from the parameters of the mixture distributions[29]. Ultimately, there is scope for alternative methods to be applied to the discovery of interesting single-cell state changes.

In existing comparison studies of scRNA-seq differential detection methods[6,30,31], analyses were limited to comparing groups of cells and had not explicitly considered sample-level inferences or aggregation approaches. The rapid uptake of new single-cell technologies has driven the collection of scRNA-seq datasets across multiple samples. Thus, it remains to be tested whether existing methods designed for comparing expression in scRNA-seq data are adequate for such cross-sample comparisons, and in particular, how sensitive aggregation methods are to detect subpopulation-level responses.

In this study, we developed a simulation framework, which is anchored to a reference dataset, that mimics various characteristics of scRNA-seq data and used it to evaluate 16 DS analysis methods (see Supplementary Table 1) across a wide range of simulation scenarios, such as varying the number of samples, the number of cells per subpopulation, and the magnitude and type of differential expression pattern introduced. We considered two conceptually distinct representations of the data for each subpopulation, cell-level or sample-level, and from these, made sample-level inferences. On cell-level data, we applied: (i) MM with a fixed effect for the experimental condition and a random effect for sample-level variability; (ii) approaches comparing full distributions (e.g., K-sample Anderson–Darling test[32]); and, as a reference point, we applied well-known scRNA-seq methods, such as scDD[33] and MAST[34], although these methods were not specifically intended for the across-sample situation. Alternatively, we assembled sample-level data by aggregating measurements for each subpopulation (for each sample) to obtain pseudobulk data in several ways; we then leveraged established bulk RNA-seq analysis frameworks to make sample-level inferences.

All methods tested are available within the *muscat* R package and a *Snakemake*[35] workflow was built to run simulation replicates. Since the discovery of state changes in cell subpopulations is an open area of research, anchor datasets are openly available via Bioconductor's *ExperimentHub*, to facilitate further bespoke method development.

Using existing pipelines for integrating, visualizing, clustering, and annotating cell subpopulations from a replicated multicondition

dataset of the mouse cortex, we applied pseudobulk DS analysis to unravel subpopulation-specific responses within brain cortex tissue from mice treated with lipopolysaccharide (LPS).

## Results

**Simulation framework**. To explore the various aspects of DS analysis, we developed a straightforward but effective simulation framework that is anchored to a labeled multi-sample multi-subpopulation scRNA-seq reference dataset and exposes parameters to modulate: the number of subpopulations and samples simulated, the number of cells per subpopulation (and sample), and the type and magnitude of a wide range of patterns of differential expression. Using (nonzero-inflated) negative binomial (NB) as the canonical distribution for droplet scRNA-seq datasets[6,36], we first estimate subpopulation- and sample-specific means, dispersion, and library size parameters from the reference data set (see Fig. 1a). Baseline multi-sample simulated scRNA-seq data can then be simulated also from an NB distribution, by sampling from the subpopulation/sample-specific empirical distributions of the mean, dispersion, and library size. To this baseline, genes can be selected as subpopulation-specific (i.e., mean different in one subpopulation vs. the others), or as a state gene (differential expression introduced in the samples from one condition), or neither (equal relative expression across all samples and subpopulations). To introduce changes in expression that represent a change in cell state, we follow the differential distribution approach of Korthauer et al.[33], adding changes in the mean expression (DE), changes in the proportions of low and high expression-state components (DP), differential modality (DM), or changes in both proportions and modality (DB). Genes that are not subject to state changes are either equivalently expressed (EE), or expressed at low and high expression-states by an equal proportion (EP) of cells in both conditions; see Fig. 1b. Here, the changes are added to samples in a condition-specific manner, thus mimicking a subpopulation-specific state change amongst replicates of one condition.

As reference datasets, we used (i) scRNA-seq data of peripheral blood mononuclear cells (PBMCs) from eight lupus patients measured before and after 6 h-treatment with IFN-$\beta$ (16 samples in total)[20], where cells were already annotated into various immune subpopulations; and, (ii) single-nuclei RNA-seq data of brain cortex tissue from eight mice split into a vehicle and LPS treatment group. In order to introduce known state changes, simulations were based only on control and vehicle samples, respectively. Importantly, our simulation framework is able to reproduce important characteristics of individual scRNA-seq datasets (e.g., mean-dropout and mean-variance relationships) from a *countsimQC*[37] analysis (see Supplementary File 1) as well as sample-to-sample variability, as illustrated by pseudobulk-level dispersion-mean trends (Supplementary Fig. 1a). By varying the proportion of subpopulation-specific and DS genes, we are able to generate multiple subpopulations that are distinct but proximal, and clearly separated from one another in lower-dimensional space (Fig. 1c); in particular, parameters control the distinctness of each subpopulation and of the group-wise state changes. Subpopulation-specific log-fold-changes (logFCs) further allow modulating differential expression to be of equal magnitude across all subpopulations, or such that a given subpopulation exhibits a weakened (logFC < 2), amplified (logFC > 2), or null (logFC = 0) differential signal relative to the default (logFC = 2; see Fig. 1c). Taken together, we constructed a simulation that replicates aspects of individual scRNA-seq datasets, mimics sample-to-sample variability, and offers a high level of flexibility to introduce subpopulation-specific identities (e.g., via marker genes) as well as condition-specific state changes.

**Aggregation vs. non-aggregation methods**. The starting point for a DS analysis is a (sparse) matrix of gene expression, either as counts (with library or size factors) or normalized data (log-transformed expression values, residuals[38,39]), where each row is a gene and each column a cell. Each cell additionally has a subpopulation (cluster) label as well as a sample label; metadata should be linked to samples, such that they can be organized into comparable groups with sample-level replicates (e.g., via a design matrix). The data processing aspect, depending on whether to aggregate data to the subpopulation-sample level, is described in the schematic in Fig. 1d. The methods presented here are modular and thus the subpopulation label could originate from an earlier step in the analysis, such as clustering[40–42] after integration[9,43] or after inference of cell-type labels at the subpopulation-[10] or cell-level[11]. The specific details and suitability of these various preprocessing steps are an active area of current research and a full evaluation of them is beyond the scope of the current work; a comprehensive review was recently made available[44].

For aggregation-based methods, we considered various combinations of input data (log-transformed expression values, residuals, counts), summary statistics (mean, sum), and methods for differential testing (*limma-voom*, *limma-trend*, *edgeR*) that are sensible from a methodological perspective. For example, *limma-voom* and *edgeR* operate naturally on pseudobulk counts, while we have also used *limma-trend* on the mean of log-transformed library-size-normalized counts (log counts). *MAST*[34] was run on log counts; AD tests[32] and *scDD*[33] on both log counts and standardized residuals (vstresiduals)[38]. For the AD tests, we considered two distinct approaches to test for equal distributions, with alternative hypotheses having samples different either sample-wise or group-wise (see Supplementary Table 1 and Methods).

**Performance of DS detection**. First, we generated null simulations where no genes are truly differential (across conditions), to evaluate the ability of methods to control error rates (3 replicates in each of 2 conditions, $K = 2$ subpopulations). While various methods show mild departures from uniform (Supplementary Fig. 2a), the AD tests, regardless of whether they were run comparing groups or samples, deviated the furthest from a uniform and were the most unstable across replicates.

To compare the ability of methods to detect DS genes, we simulated $S_1 = S_2 = 3$ samples across 2 conditions. To retain the empirical distribution of library sizes, we simulated the same number of genes as in the reference dataset and selected a random subset of $G = 4000$ genes for further analysis to reduce runtimes. We simulated $K = 3$ subpopulations and introduced 10% of genes with DS, with an equal magnitude of differential expression across subpopulations ($\mathbb{E}[\mathrm{logFC}] = 2$) and randomly assigned to genes across the range of expression strength.

To ensure that method performances are comparable and do not suffer from low cell numbers, we simulated an average of 200 cells per subpopulation-sample instance, amounting to a total of $\sim 200 \times (S_1 + S_2) \times K \approx 3,600$ cells per simulation. Each simulation and method was repeated five times per scenario, and performances were averaged across replicates.

In the context of DS analysis, each of the $G$ genes is tested independently in each of $K$ subpopulations, resulting in a total of $\sim G \times K$ differential tests (occasionally, a small number of genes are filtered out due to low expression). Multiple testing correction could thus, in principle, be performed globally, i.e., across all tests ($n = G \times K$), or locally, i.e., on each of the subpopulation-level tests ($n = G$). We compared overall false-discovery rate (FDR) and true positive rate (TPR) estimates computed from both

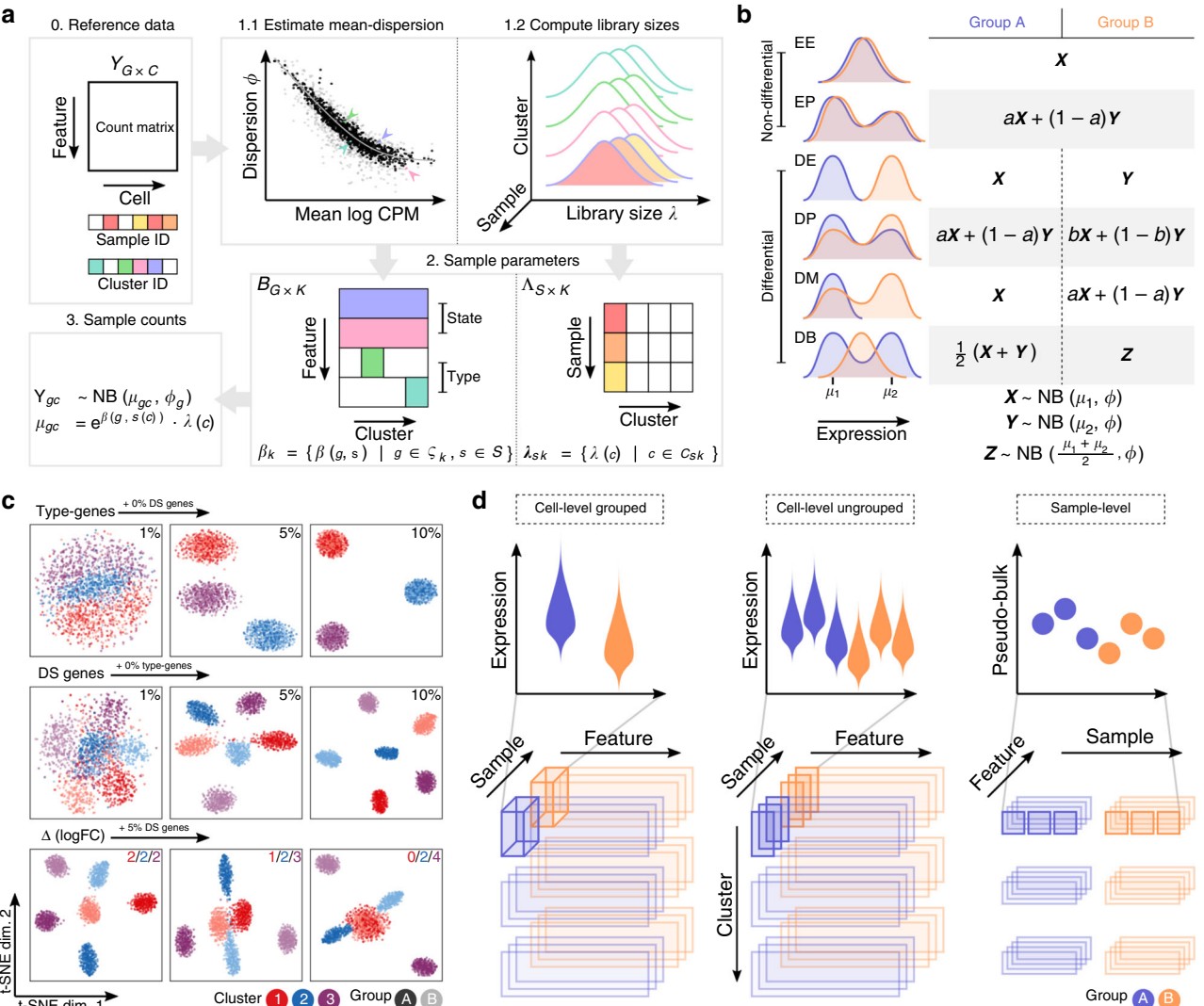

**Fig. 1 Schematic overview of *muscat*'s simulation framework. a** Given a count matrix of features by cells and, for each cell, predetermined cluster (subpopulation) identifiers as well as sample labels (0), dispersion and sample-wise means are estimated from a negative binomial distribution for each gene (for each subpopulation) (1.1); and library sizes are recorded (1.2). From this set of parameters (dispersions, means, and library sizes), gene expression is sampled from a negative binomial distribution. Here, genes are selected to be "type" (subpopulation-specifically expressed; e.g., via marker genes), "state" (change in expression in a condition-specific manner), or equally expressed (relatively) across all samples (2). The result is a matrix of synthetic gene expression data (3); **b** Differential distributions are simulated from an NB distribution or mixtures thereof, according to the definitions of random variables *X*, *Y*, and *Z*. **c** t-SNE plots for a set of simulation scenarios with varying percentage of "type" genes (top), DS genes (middle), and the difference in the magnitude (logFC) of DS between subpopulations (bottom). **d** Schematic overview of cell- and sample-level approaches for DS analysis. Top panels show a schematic of the data distributions or aggregates across samples (each violin is a group or sample; each dot is a sample) and conditions (blue or orange). The bottom panels highlight the data organization into sub-matrix slices of the original count table.

locally and globally adjusted *p* values. Global *p* value adjustment led to a systematic reduction of both FDRs and TPRs (Fig. 2a; stratified also by the type of DS) and is therefore very conservative.

Moreover, detection performance is related to expression level, with differences in lowly expressed genes especially difficult to detect (Supplementary Fig. 4). On the basis of these observations, for the remainder of this study, all method performances were evaluated using locally adjusted *p* values, after the exclusion of genes with a simulated expression mean below 0.1.

In general, all methods performed best for the genes of the DE category, followed by DM, DP, and DB (Fig. 2a). This level of difficulty by DS type is to be expected, given that genes span the range of expression levels and imposing mixtures of expression changes (DM and DP) dampens the overall magnitude of change

compared to DE. In particular, DB, where the means are not different in the two conditions, is particularly difficult to detect, especially at low expression; therefore, several methods, including most of those that analyze full distributions (AD, *scDD*), underperform in this situation. For example, the AD tests on vstresiduals show good sensitivity but also result in unacceptably high FDRs. For DE, DM, and DP, there is a set of methods that perform generally well, including most of the pseudobulk approaches and cell-level MM models. Aggregation- and MM-based methods also performed fairly consistent across simulation replicates, while other methods were generally more erratic in their performance (Supplementary Fig. 3).

Comparison of simulated and estimated logFC highlighted that MM-based methods and *limma-trend* applied to mean-log counts systematically underestimate logFCs, with estimates falling close

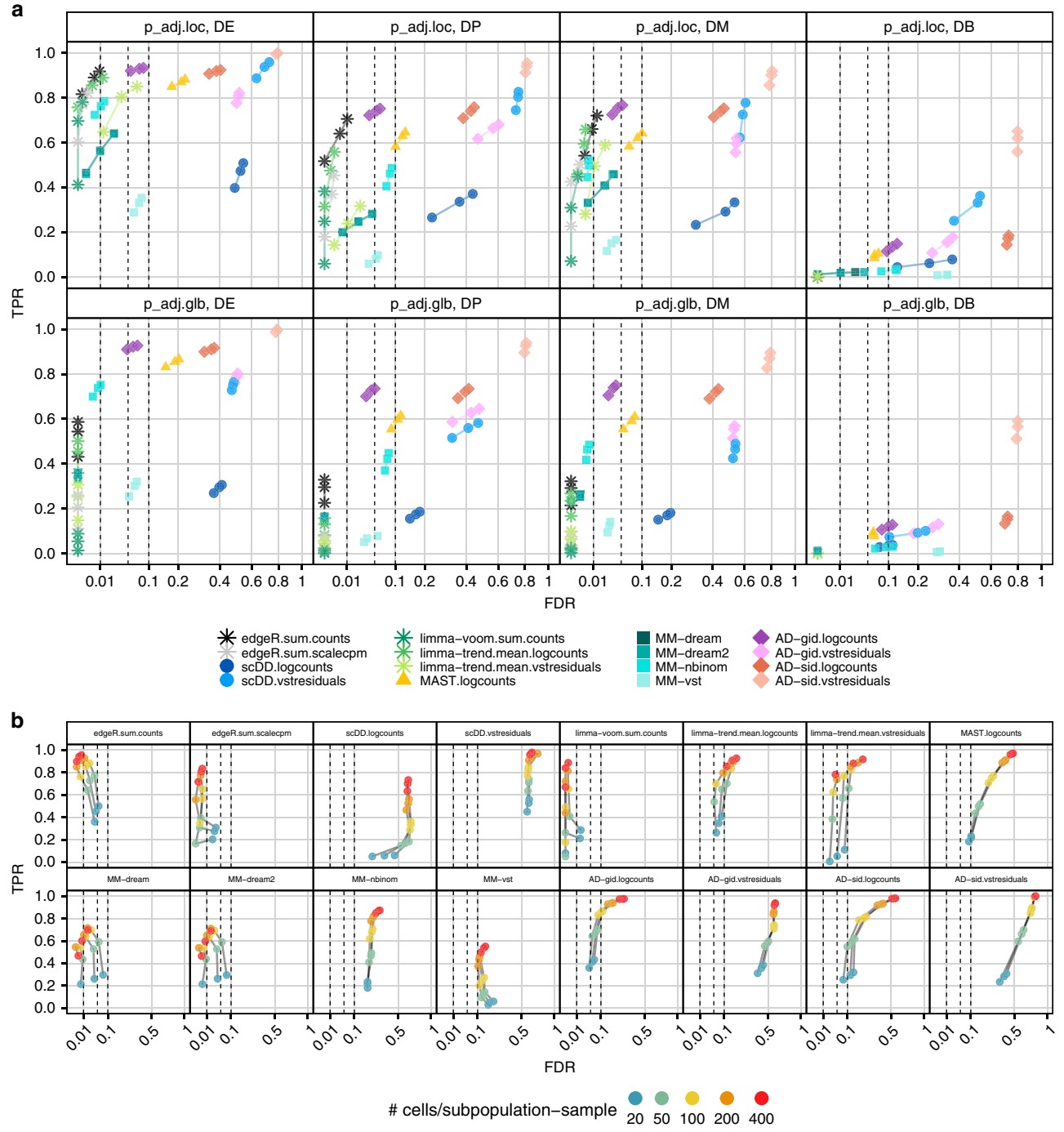

**Fig. 2 DS method performance across *p* value adjustment types, *differential distribution* categories, and subpopulation-sample cell counts.** All panels show observed overall true positive rate (TPR) and false discovery rate (FDR) values at FDR cutoffs of 1%, 5%, and 10%; dashed lines indicate desired FDRs (i.e., methods that control FDR at their desired level should be left of the corresponding dashed lines). For each panel, performances were averaged across five simulation replicates, each containing 10% of DS genes (of the type specified in the panel labels of (**a**), and 10% of DE genes for (**b**); see Fig. 1b for further details). **a** Comparison of locally and globally adjusted *p* values, stratified by DS type. Performances were calculated from subpopulation-level (locally) adjusted *p* values (top row) and cross-subpopulation (globally) adjusted *p* values (bottom row), respectively. **b** Performance of detecting DS changes according to the number of cells per subpopulation-sample, stratified by the method.

to zero for a large fraction of gene-subpopulation combinations (Supplementary Fig. 5a). Although the differential detection performance does not seem to be compromised, applying the logarithm transformation (with an offset to avoid zero) to the rather low counts of cell-level data attenuates the scale and thus the magnitude of the estimated logFCs. For the remainder of

methods, simulated and estimated logFC showed high correspondence across all gene categories.

To investigate the effect of subpopulation size on DS detection, we ran methods on simulations containing 10% of DE genes using subsets of 20–400 cells per subpopulation-sample (Fig. 2b). For most methods, FDR control varies drastically with the

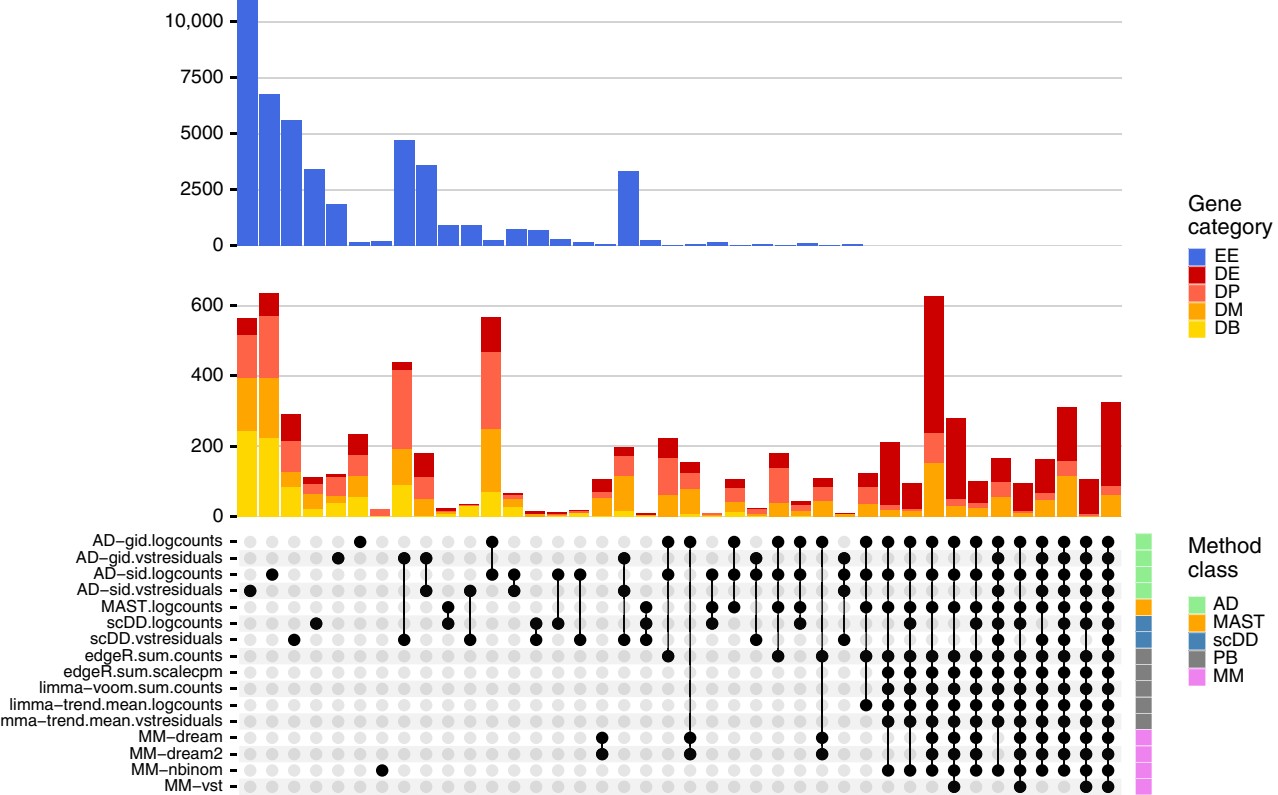

**Fig. 3 Between-method concordance.** Upset plot obtained from intersecting the top-*n* ranked gene-subpopulation combinations (lowest *p* value) across methods and simulation replicates. Here, $n = \min(n_1, n_2)$, where $n_1 =$ number of genes simulated to be differential, and $n_2 =$ number of genes called differential at FDR < 0.05. Shown are the 40 most frequent interactions; coloring corresponds to (true) simulated gene categories. The bottom right annotation indicates method types (PB pseudobulk (aggregation-based) methods, MM mixed models, and AD Anderson–Darling tests).

number of cells, while TPRs improve for more cells across all methods. For aggregation-based methods, ~100 cells were sufficient to reach decent performance; in particular, there is a sizable gain in performance in going from 20 to 100 cells (per subpopulation per sample), but only a moderate gain in a deeper sampling of subpopulations (e.g., 200 or 400 cells per subpopulation per sample). Except for *edgeR* on pseudobulk summed scaled CPM, unbalanced sample and group sizes had no effect on method performances (Supplementary Figs. 6 and 7), and increasing the number of replicates per group reveals the expected, although modest, increase in detection performance (Supplementary Fig. 8).

To investigate the overall method concordance, we intersected the top-ranked DS detections (FDR < 0.05) returned by each method across five simulation replicates per DS category (Fig. 3). We observed overall high concordance between methods, with the majority of common hits being truly differential. In contrast, most isolated intersections, i.e., hits unique to a certain method, were genes that had been stimulated to be EE and thus false discoveries. Methods with vstresiduals as input yielded a noticeably high proportion of false discoveries.

Using a different anchor dataset as input to our simulation framework yielded highly consistent results (Supplementary Figs. 1b, 2b, 3b, 5b, 6b, 9, 10, and Supplementary File 2). Method runtimes varied across several orders of magnitude (Supplementary Fig. 11). MMs were by far the slowest, followed by AD tests, *MAST*, and then *scDD*. Aggregation-based DS methods were the fastest. *MAST*, *scDD*, and MMs provide arguments for parallelization, and all methods could be implemented to parallelize computations across subpopulations. For comparability, all methods were run here on a single core.

**DS analysis of mouse cortex exposed to LPS treatment**. One of the motivating examples for the DS methodological work was a scRNA-seq dataset collected to understand how peripheral LPS induces its effects on brain cortex. LPS given peripherally is capable of inducing a neuroinflammatory response. Even if the mechanisms at the base of this response are still not clear, it is known that LPS can penetrate the blood–brain barrier (BBB) or alternatively, can act outside the BBB by stimulating afferent nerves, acting at circumventricular organs, and altering BBB permeabilities and functions[45–48].

We sought to investigate the effects of peripheral LPS administration on all major cell types in the mouse frontal cortex using single-nuclei RNA-seq (snRNA-seq). The goal was to identify genes and pathways affected by LPS in neuronal and non-neuronal cells. To this end, we applied our DS analysis framework to snRNA-seq data of four control (vehicle) and four LPS-treated mice using pseudobulk (sum of counts) and *edgeR*. We obtained 12,317 vehicles and 12,907 treated cells that passed filtering and received a subpopulation assignment. Using graph-based clustering (Louvain algorithm[49]), we identified 22 cell clusters and annotated them into 8 subpopulations (using both canonical and computationally identified marker genes): astrocytes, endothelial cells, microglia, oligodendrocyte progenitor cells (OPC), choroid plexus ependymal (CPE) cells, oligodendrocytes, excitatory neurons, and inhibitory neurons (see "Methods" and Supplementary File 3). Low dimensional projections of cells and pseudobulks (by subtype and condition) are shown in Figs. 4a through c; sample sizes and relative subpopulation abundances are shown in Supplementary Fig. 12.

We identified 915 genes with DSs (FDR < 0.05, |logFC| > 1) in at least one subpopulation, 751 of which were detected in only a

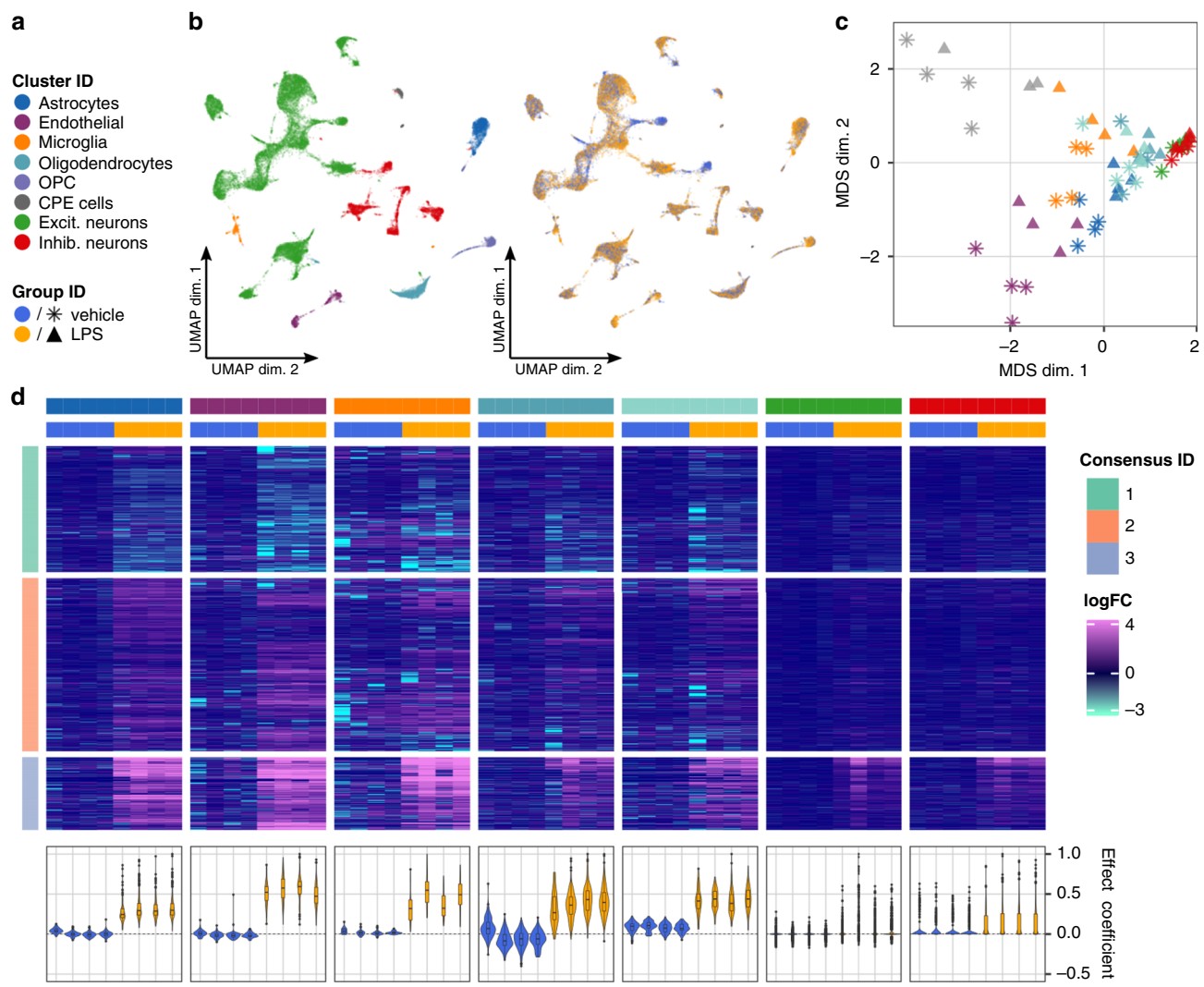

**Fig. 4 DS analysis of cortex tissue from vehicle- and LPS-treated mice. a** Shared color and shape legend of subpopulation and group IDs. **b** UMAP visualization colored by subpopulation (left) and group ID (right). **c** Pseudobulk-level Multidimensional Scaling (MDS) plot. Each point represents one subpopulation-sample instance; points are colored by subpopulation and shaped by group ID. **d** Heatmap of pseudobulk-level log-expression values normalized to the mean of vehicle samples; rows correspond to genes, columns to subpopulation-sample combinations. Included is the union of DS detections (FDR < 0.05, |logFC| > 1) across all subpopulations. Data are split horizontally by subpopulation and vertically by consensus clustering ID (of genes); top and bottom 1% logFC quantiles were truncated for visualization. Bottom-row violin plots represent cell-level effect coefficients computed across all differential genes, and scaled to a maximum absolute value of 1 (each violin is a sample; coloring corresponds to group ID); effect coefficients summarize the extent to which each cell reflects the population-level fold-changes (see "Methods").

single subpopulation (Supplementary Fig. 13). Since relying on thresholds alone is prone to bias, we next clustered the (per-subpopulation) fold-changes across the union of all differentially expressed genes (Fig. 4d). We observed a distinct set of genes (consensus clustering ID 3) that were upregulated across all subpopulations and enriched for genes associated with response to (external) biotic stimulus, defense, and immune response (Supplementary File 4). Endothelial cells appeared to be most strongly affected, followed by glial cells (astrocytes, microglia, and oligodendrocytes). While the responses for consensus cluster 3 were largely consistent across all subpopulations, some genes' responses departed from the trend (e.g., are specific to a single subpopulation or subset of subpopulations (Supplementary Fig. 14).

We next sought to estimate how homogeneous the effects observed at the pseudobulk-level are across cells. To this end, we calculated effect coefficients summarizing the extent to which each cell reflects the population-level fold-changes (Fig. 4d,

bottom). For endothelial and glial cells, the effect coefficient distributions were well separated between vehicle and LPS samples, indicating that the majority of cells are affected. In contrast, the large overlap of the distributions in neurons suggests that only a minority of cells react. Taken together, these analyses clearly demonstrate the ability of our DS analysis framework to identify and characterize subpopulation-specific as well as global state transitions across experimental conditions.

In order to investigate the concordance of the 16 surveyed DS methods on a real dataset, we applied all methods to the LPS dataset. Intersecting genes reported as differential (at FDR < 0.05) yielded results similar to the simulation study (Supplementary Fig. 15); for example, AD, MAST, and scDD methods report large numbers of isolated hits, whereas overall high agreement between aggregation- and MM-based methods is observed. While formal evaluation of method performance is not possible in the absence of ground truth, these results reveal nonetheless similar trends to the simulation results.

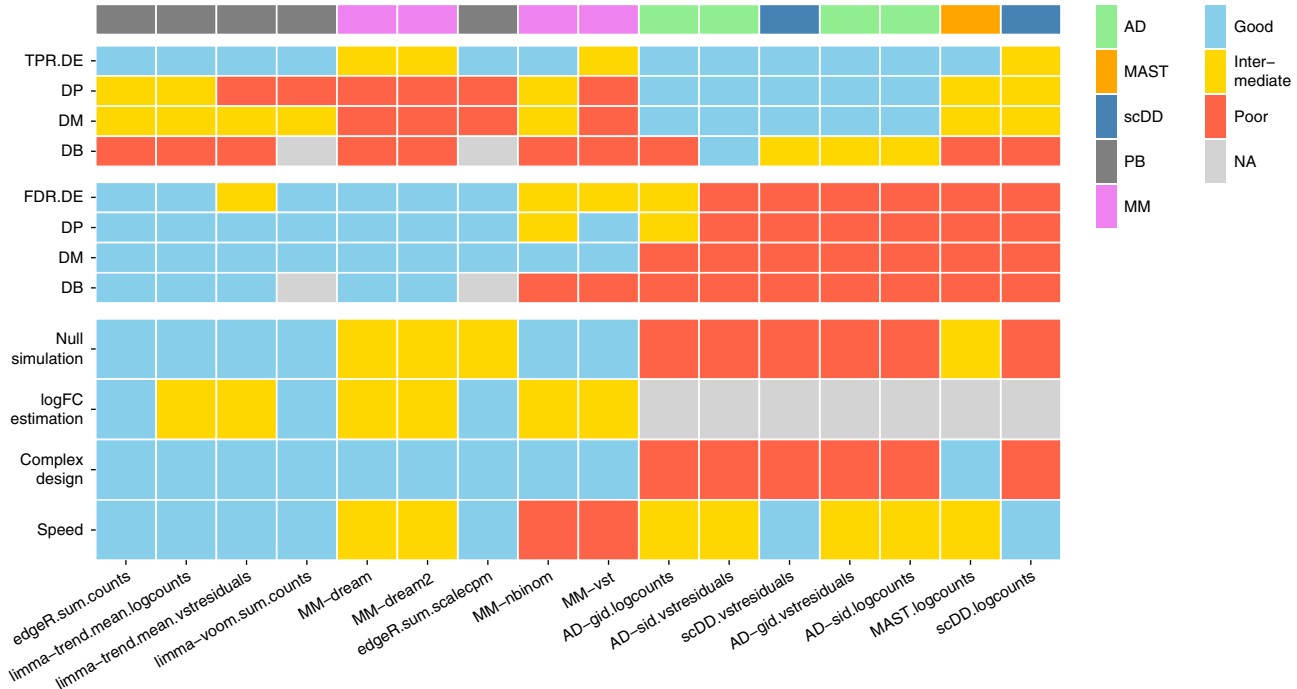

**Fig. 5 Summary of DS method performance across a set of evaluation criteria.** Methods are ranked from left to right by their weighted average score across criteria, with the numerical encoding good = 2, intermediate = 1, and poor/NA = 0. Evaluation criteria (y-axis) comprise DS detection sensitivity (TPR) and specificity (FDR) for each type of differential distribution, uniformity of p value distributions under the null (null simulation), concordance between simulated and estimated logFCs (logFC estimation), ability to accommodate complex experimental designs (complex design), and runtimes (speed). Top annotation indicates method types (PB pseudobulk (aggregation-based) methods, MM mixed models, AD Anderson–Darling tests). Null simulation, logFC estimation, complex design, and runtimes received equal weights of 0.5; TPR and FDR were weighted according to the frequencies of modalities in scRNA-seq data reported by Korthauer et al.[33]: ~75% unimodal, ~5% trimodal, and ~25% bimodal, giving weights of 0.75 for DE, 0.125 for DP and DM, and 0.05 for DB.

## Discussion

We have compared what can be considered as *in silico sorting* approaches for multi-subpopulation multi-sample multi-condition scRNA-seq datasets, where the interest is to follow each cell subpopulation along the axis of samples and conditions; we refer to these generally as DS analyses and have largely leveraged existing tools for running such analyses. A summary of the tested DS methods across several criteria (e.g., sensitivity and runtimes) is given in Fig. 5; methods were scored quantitatively and partially on visual inspection of the simulation results (see Methods). Furthermore, we have applied DS analysis to a new dataset to uncover subpopulation-specific changes in brain tissue from mice exposed to peripheral LPS treatment.

Aggregating data from a subpopulation to a single observation (per sample) is a natural approach to the DS problem[20,21], but it still remained to be demonstrated how effective it is. Based on our simulation results, the tested aggregation-based DS methods were extremely fast and showed overall a stable high performance, although depending on the scale of the data analyzed, logFCs were attenuated for some combinations. While MM methods performed similarly well, their computational cost may not be worth the flexibility they provide (Fig. 5 and Supplementary Fig. 11). Methods developed specifically for scRNA-seq differential analysis were outperformed by aggregation and MMs, but it should be mentioned that these methods focus on comparing sets of cells and were not specifically designed for the multi-group multi-sample problem. Furthermore, methods that compared full distributions did not perform well overall (Fig. 5). This latter class of methods was used here as a reference point, but could also be improved to be more targeted to the DS inference problem. For

example, AD tests were run in two ways, group-wise or sample-wise, where under the null hypothesis, all distributions are equal. In the sample-wise case, departures from the null could happen between replicates of the same experimental condition, and in the group-wise case, it is perhaps not ideal to mix distributions from different samples. Thus, while our results suggest that aggregation methods are fast and perform amongst the best, there may still be value in considering full distributions, if bespoke methods were developed. Furthermore, methods that integrate both changes in the mean and changes in variability may be worth exploring.

The starting point of a DS analysis is a count table across genes and cells, where each cell has an appropriate subpopulation and sample label, and metadata (e.g., patient, experimental condition information) accompanies the list of samples. This starting point, organization of cells into subpopulations ("types"), is itself an active and debated area of research[2,3] and one that already applies a computational analysis on a given dataset, whether that be clustering or manual or computational assignment; in fact, combining computational and manual assignment was recently listed as best practice[44].

Although not discussed here, researchers would generally first apply a differential abundance (DA) analysis of subpopulations, which naturally leads to discussions about the ambiguity of cell type definitions. DA analysis will highlight subpopulations whose relative abundance changes according to the treatment; in contrast, DS analysis will identify changes within the defined subpopulations that are associated with the treatment. Thus, the combination of DA and DS should always be capable of detecting interesting differences, with results dependent on how cell types are defined.

Another aspect of subpopulation-level analyses is that there are clear connections to existing tools and practices in the analysis of gene expression. For example, one can visualize data at the aggregate level (e.g., MDS plot for each subpopulation; Fig. 4c) and apply standard tools (e.g., geneset analysis, gene network analysis) for discovery and interpretation on each subpopulation, thus leveraging existing methods.

By default, we have focused on subpopulation-specific DS analysis; in particular, the methods fit a separate model (i.e., separate dispersion) for each subpopulation, which explicitly allows them to have different levels of variability. However, some of the models could be reshaped, e.g., to fit a single model overall subpopulations and test parameters within this model. This strategy may allow better separation of features that respond globally versus specific to a given subpopulation, which may be important to separate in the downstream interpretation analyses.

The number of cells required to detect DS changes depends on many factors, including the effect's strength, the number of replicates, the number of cells per sample in each subpopulation, and the sensitivity of the scRNA-seq assay, which itself is a moving target. In general, there is a clear gain in power for larger subpopulations, while FDR control can vary greatly with the number of cells (Fig. 2). Going forward, it would be of interest to further explore the origins of this instability, in order to better maximize sensitivity while still controlling for errors.

Another aspect to consider in this context is the resolution of subpopulations subjected to differential testing; for example, there is an analogous tradeoff between sensitivity (e.g., larger subpopulations) and specificity (e.g., effects that target particular subpopulations). Here, methods that integrate the relationship between subpopulations (e.g., *treeclimbR*[50]) could be applied as an additional layer to improve signal detection.

In the process of this study, we created a flexible simulation framework to facilitate method comparisons as well as data handling tools and pipelines for such experiments, implemented in the *muscat* R package. By using sample-specific estimates, inter-sample variability present in the reference dataset will be represented in the simulated data. Even though we tested here a broad set of scenarios, there may be other scenarios of interest (e.g., different percentages of the DM mixtures); the simulation framework provided in the *muscat* package could readily be used to expand the set of simulation scenarios. Furthermore, the simulation framework could be extended to induce batch effects via, for example, incorporating sample-specific logFCs in the computation of simulation means. For this, more research needs to be done to understand how and at what magnitude batch effects manifest. Furthermore, our simulation framework could be extended: (i) to accommodate an arbitrary number of groups for which the magnitude of the differential signal, the percentage of differential genes, as well as the set of affected subpopulations could be varied; or, (ii) implementing *type* genes such that they are not specific to a single subpopulation, perhaps even in a hierarchical structure to represent markers of both broad and specific cell types. Taken together, we expect our simulation framework to be useful to investigate various scRNA-seq data analyses, such as batch correction frameworks, clustering, reference-based cell-type inference methods, marker gene selection methods as well as further developments in DS analysis.

Although we set out with the goal of discovering subpopulation-specific responses across experimental conditions, one needs to be careful in how strongly these claims are made. The absence of evidence is not evidence of absence. In particular, there is a potentially strong bias in statistical power to detect changes in larger cell populations, with decreased power for rarer populations. Statistical power to detect changes in cell states

also relates to the depth of sequencing per cell; for example, it has been speculated that cell states are a *secondary* regulatory module[3] and it is unclear at this stage whether we are sequencing deeply enough to access all of the interesting transcriptional programs that relate to cell state. However, despite the potential loss of single-cell resolution, aggregation approaches should be helpful in this regard, accessing more genes at the subpopulation level.

## Methods

**Preprocessing of simulation reference data**. As simulation anchors, we used scRNA-seq datasets obtained from (i) PBMCs by Kang et al.[20] (eight control vs. eight IFN-$\beta$ treated samples); and, (ii) mouse brain cortex cells (four vehicles vs. four LPS-treated samples; see below). In order to introduce known changes in expression, we only used samples from the reference (control and vehicle, respectively) condition as an input to our simulation framework. These were minimally filtered to remove cells with less than 200 detected genes, and genes detected in less than 100 cells. Available metadata was used to filter for singlet cells as well as cells that have been assigned to a cell population. Finally, for more accurate parameter estimation, only subpopulation-sample instances with at least 100 cells were retained, leaving 4 samples per reference dataset, 4 subpopulations for the Kang et al.[20], and 3 subpopulations for the LPS dataset.

**Simulation framework**. The simulation framework (Fig. 1a) comprises (i) estimation of NB parameters from a reference multi-subpopulation, multi-sample dataset; (ii) sampling of gene and cell parameters to use for simulation; and, (iii) simulation of gene expression data as negative binomial (NB) distributions or mixtures thereof.

Let $Y = (y_{gc}) \in \mathbb{N}_0^{G \times C}$ denote the count matrix of a multi-sample multi-subpopulation reference dataset with genes $\mathcal{G} = \{g_1, \ldots, g_G\}$ and sets of cells $\mathcal{C}_{sk} = \{c_1^{sk}, \ldots, c_{C_{sk}}^{sk}\}$ for each sample $s$ and subpopulation $k$ ($C_{sk}$ is the number of cells for sample $s$, subpopulation $k$). For each gene $g$, we fit a model to estimate sample-specific means $\beta_g^s$, for each sample $s$, and dispersion parameters $\phi_g$ using edgeR's *estimateDisp* function with default parameters. Thus, we model the reference count data as NB distributed: $Y_{gc} \sim \text{NB}(\mu_{gc}, \phi_g)$ for gene $g$ and cell $c$, where the mean $\mu_{gc} = \exp(\beta_g^{s(c)}) \cdot \lambda_c$. Here, $\beta_g^{s(c)}$ is the relative abundance of gene $g$ in sample $s(c)$, $\lambda_c$ is the library size (total number of counts), and $\phi_g$ is the dispersion.

In order to introduce a multi-subpopulation, multi-sample data structure, we sample a set of $K$ clusters as a reference, as well as $S$ reference samples for each of the two groups, resulting in an unpaired design. Alternatively, the pairing of samples can be mimicked by fixing the same set of reference samples for both groups. For each subpopulation $k \in \{1, \ldots, K\}$, we sample a set of genes $\mathcal{G}_k^* \subset \mathcal{G}$ used for simulation, such that most genes are common to all subpopulations ($\mathcal{G}_1^* \cap \mathcal{G}_2^* \cap \ldots \cap \mathcal{G}_K^* \approx (1 - p) \cdot G$), while a small set ($p \cdot 100\%$) of *type*-specific genes are sampled separately for each subpopulation ($\mathcal{G}_k \cap \mathcal{G}_{k'} = \emptyset \; \forall \; k \neq k'$), giving rise to distinct subpopulations. Secondly, for each sample $s$ and subpopulation $k$, we draw a set of cells $\mathcal{C}_{sk}^* \subset \mathcal{C}_{sk}$ (and their corresponding $\lambda_c$, $\beta_g^{s(c)}$, and $\phi_g$) to simulate (NB random variables) from, where cells $\mathcal{C}_{sk}$ belong to the corresponding reference cluster-sample drawn previously.

Lastly, the differential expression of a variety of types is added for a subset of genes. For each subpopulation, we randomly assign each gene to a given *differential distribution* category according to a probability vector ($p_{EE}, p_{EP}, p_{DE}, p_{DP}, p_{DM}, p_{DB}$); see Fig. 1b. For each gene and subpopulation, we draw a vector of fold changes from a Gamma distribution with shape 4 and rate 4/$\mu_{\text{logFC}}$, where $\mu_{\text{logFC}}$ is the desired average logFC across all genes and subpopulations. The direction of differential expression is randomized for each gene, with an equal probability of up- and down-regulation. We split the cells in a given subpopulation-sample combination into two sets (representing treatment groups), $\mathcal{T}_A$ and $\mathcal{T}_B$, which are in turn split again into two sets each (representing subpopulations within the given treatment group), $\mathcal{T}_{A_1}/\mathcal{T}_{A_2}$ and $\mathcal{T}_{B_1}/\mathcal{T}_{B_2}$.

For EE genes, counts for $\mathcal{T}_A$ and $\mathcal{T}_B$ are drawn using identical means. For EP genes, we multiply the effective means for identical fractions of cells per group by the sampled FCs, i.e., cells are split such that $\dim \mathcal{T}_{A_1} = \dim \mathcal{T}_{B_1}$ and $\dim \mathcal{T}_{A_2} = \dim \mathcal{T}_{B_2}$. For DE genes, the means of one group, $A$ or $B$, are multiplied with the sampled FCs. DP genes are simulated analogously to EP genes with $\dim \mathcal{T}_{A_1} = a \cdot \dim \mathcal{T}_A$ and $\dim \mathcal{T}_{B_1} = b \cdot \dim \mathcal{T}_B$, where $a + b = 1$ and $a \neq b$ (default $a = 0.3$, $b = 0.7$). For DM genes, 50% of cells from one group are simulated at $\mu \cdot$ FC. For DB genes, all cells from one group are simulated at $\mu \cdot \frac{\text{FC}}{2}$, and the second group is split into EPs of cells simulated at $\mu$ and $\mu \cdot$ FC, respectively.

Details on all simulation parameters, illustrative examples of their effects, and instructions on how to generate an interactive quality control report and benchmark DS methods through simulated data are provided in the *muscat* R/Bioconductor package's documentation (see Software specification and code availability).

**Aggregation-based methods**. We summarize the input measurement values for a given gene over all cells in each subpopulation and by sample. The resulting pseudobulk data matrix has dimensions $G \times S$, where $S$ denotes the number of samples, with one matrix obtained per subpopulation. Depending on the specific method, which includes both a type of data to operate on (e.g., counts, log counts) and summary function (e.g., mean, sum), the varying number of cells between samples and subpopulations is accounted for prior to or following aggregation. For log counts methods, we apply a library size normalization to the input raw counts. vstresiduals are computed using R package *sctransform*'s *vst* function[38]. For sca-lecpm, we calculate the total library size of each subpopulation $k$ and sample $s$ as:

$$\Lambda_{sk} = \sum_{g=1}^{G} \sum_{c=1}^{C_{sk}} y_{gc},$$

where $G$ represents the number of genes, $C_{sk}$ is the total number of cells in sample $s$ that have been assigned to subpopulation $k$, and $y_{gc}$ denotes the counts observed for gene $g$ in cell $c$. We then multiply the CPM of a given sample and subpopulation with the respective total library size in millions to scale the CPM values back to the counting scale:

$$\mathrm{CPM}_{sk}^* = \mathrm{CPM}_{sk} \cdot \Lambda_{sk} \cdot 1e^{-6},$$

*edgeR*-based methods were run using *glmQLFit* and *glmQLFTest*[51]; methods based on *limma-voom* and *limma-trend* were run using default parameters.

**Mixed models**. MM methods were implemented using three main approaches: (i) fitting linear MMs (LMMs) on log-normalized data with observational weights, ii) fitting LMMs on variance-stabilized data, iii) fitting generalized LMMs (GLMMs) directly on counts. Subpopulations with less than 10 cells in any sample and genes detected in fewer than 20 cells were excluded from differential testing. In each case, a ~1 + group_id + (1|sample_id) model was fit for each gene, optimizing the restricted maximum likelihood (i.e., *REML=TRUE*), and *p* values were calculated using Satterthwaite estimates of degrees of freedom (the Kenward-Roger approach being longer to compute and having a negligible impact on the final results). Fitting, testing and moderation were applied subpopulation-wise.

For the first approach (*MM-dream*), we relied on the *variancePartition*[52] package's implementation for repeated measurement bulk RNA-seq, using *voom*'s[25] precision weights as originally described but without empirical Bayes moderation and the *duplicateCorrelation*[53] step, as this was computationally intensive and had a negligible impact on the significance (as also observed previously for batch effects[21]). Method *MM-dream2* uses an updated alternative to this approach using *variancePartition*'s new weighting scheme[54] instead of *voom*.

For the second approach (*MM-vst*), we first applied the variance-stabilizing transformation globally before splitting cells into subpopulations and then fitted the model using the *lme4* package[55] directly on transformed data (and without observational weights). We then applied *eBayes* moderation as in the first approach. We tested both the variance-stabilizing transformation from the *DESeq2* package[23] and that from the *sctransform* package[38], the latter of which was specifically designed for Unique Molecular Identifier based scRNA-seq; since the latter outperformed the former (data not shown), it was retained for the main results shown here.

For the GLMM-based approach (*MM-nbinom*), we supplemented the model with an offset equal to the library size factors and fitted it directly on counts using both Poisson and NB distributions (with log-link). The Poisson-distributed model was fit using the *bglmer* function of the *blme* package, while the NB model was fit with the *glmmTMB* framework (*family = nbinom1*). As *eBayes* moderation did not improve performance on these results, it was not applied in the final implementation.

All these methods and variations thereof are available through the *mmDS* function of the *muscat* package.

**Other methods**. For AD tests, we used the *ad.test* function from the *kSamples* R package[56], which applies a permutation test that uses the AD criterion[32] to test the hypothesis that a set of independent samples arose from a common, unspecified distribution. Method AD-sid uses sample labels as grouping variables, thus testing whether any sample from any group arose from a different distribution than the remaining samples. For method AD-gid, we used group labels as grouping variables, thus testing against the null hypothesis that both groups share a common underlying distribution; with disregard of sample labels. For both methods, we require genes to be expressed in at least ten cells in a given cluster to be tested for DSs.

*scDD*[33] was run using default prior parameters and *min.nonzero = 20*, thus requiring a gene to be detected in at least 20 cells per group to be considered for differential testing in a given subpopulation. For *MAST*[34], we fit a subpopulation-level zero-inflated regression model for each gene (function *zlm*) and applied a likelihood-ratio test (function *lrTest*) to test for between-group differences in each subpopulation. Both steps were run using default parameters. AD methods and *scDD* were run on both log counts and vstresiduals; *MAST* was run on log counts only.

**Animal studies—LPS dataset**. Ethical approval for this study was provided by the Federal Food Safety and Veterinary Office of Switzerland. All animal experiments were conducted in strict adherence to the Swiss federal ordinance on animal

protection and welfare as well as according to the rules of the Association for Assessment and Accreditation of Laboratory Animal Care International (AAALAC).

CD1 male mice (Charles River Laboratories, Germany) age 11 weeks were divided into two groups with 4 animals each: a vehicle and an LPS treatment group. The LPS-treated group was given a single intraperitoneal injection of LPS from Escherichia coli O111:B4 (Sigma Aldrich, L2630) at a dose of 5 mg/kg, dissolved in 0.9% NaCl. Vehicle mice were injected with a solution of DMSO/Tween80/NaCl (10%/10%/80%). The mice were sacrificed 6 h later by anesthetizing the animals with isoflurane followed by decapitation. Brains were quickly frozen and stored at −80 °C.

**Nuclei isolation, mRNA-seq library preparation, and sequencing—LPS data-set**. Nuclei were prepared using the NUC201 isolation kit from Sigma Aldrich. Briefly, $8 \times 50$ μm sagittal sections of cortex from each animal were prepared using a microtome and placed in 200 μl of cold Nuclei Pure Lysis Buffer (Nuclei Pure Prep Nuclei isolation kit—Sigma Aldrich) with 1 M dithiothreitol (DTT) and $0.2\frac{U}{\mu l}$ SUPERase inhibitors (Invitrogen) freshly added before use. Nuclei were extracted using a glass Dounce homogenizer with Teflon pestle using 10–12 up and down strokes in lysis buffer. Totally, 360 μl of cold 1.8 M Sucrose Cushion solution was added to lysate which was then filtered through a 30 μm strainer. Totally, 560 μl of filtered solution was carefully overlayed on 200μl of Sucrose solution, and nuclei were purified by centrifugation for 45 min at 16,000*g*. The nuclei pellet was re-suspended in 50 μl cold Nuclei Pure Storage Buffer (Nuclei Pure Prep Nuclei isolation kit—Sigma Aldrich) with $0.2\frac{U}{\mu l}$ SUPERase inhibitors and centrifuged for 5 min at 500*g*. The supernatant was removed, the pellet washed again with Nuclei Pure Storage Buffer with $0.2\frac{U}{\mu l}$ SUPERase inhibitors, and centrifuged for 5 min at 500*g*. Finally, the pellet was re-suspended in 50 μl cold Nuclei Pure Storage Buffer with $0.2\frac{U}{\mu l}$ SUPERase inhibitors. Nuclei were counted using trypan blue staining on Countess II (Life technology). A total of 12,000 estimated nuclei from each sample was loaded on the 10× Single Cell B Chip.

cDNA libraries from each sample were prepared using the Chromium Single Cell 3′ Library and Gel Bead kit v3 (10× Genomics) according to the manufacturer's instructions. cDNA libraries were sequenced using Illumina Hiseq 4000 using the HiSeq 3000/4000 SBS kit (Illumina) and HiSeq 3000/4000 PE cluster kit to get a sequencing depth of 30K reads/nuclei.

**Single-nucleus RNA-seq data processing and quality control**. Paired-end sequencing reads from the eight samples were preprocessed using 10× Genomics Cell Ranger 3.0 software for sample demultiplexing, barcode processing, and single-nucleus 3′ gene counting (single nuclei mode; counting performed on unspliced Ensembl transcripts, as described in the 10x Genomics documentation). Mouse reference genome assembly GRCm38/mm10 was used for the alignment of sequencing reads. The gene by cell count matrices generated by the Cell Ranger pipeline was used for downstream quality control and analyses.

**LPS dataset analysis**. Filtering for doublet cells was performed on each sample separately using the hybrid method of the *scds* package[57], removing the expected 1% per thousand cells captured with the highest doublet score. Quality control and filtering were performed using the *scater*[58] R package. Upon removal of genes that were undetected across all cells, we removed cells whose feature counts, number of expressed features, and percentage of mitochondrial genes fell beyond 2.5 median absolute deviations of the median. Finally, features with a count >1 in at least 20 cells were retained for downstream analysis.

Next, we used *Seurat*[9,43] v3.0 for integration, clustering, and dimension reduction. Integration and clustering were performed using the 2000 most highly variable genes (HVGs) identified via *Seurat*'s *FindVariableFeatures* function with default parameters; integration was run using the first 30 dimensions of the canonical correlation analysis cell embeddings. Clusterings as well as dimension reductions (t-SNE[59] and UMAP[60]) were computed using the first 20 principal components. For clustering, we considered a range of *resolution* parameters (0.1–2); downstream analyses were performed on cluster assignments obtained from *resolution* 0.2 (22 subpopulations).

Cluster merging and cell-type annotation were performed manually on the basis of a set of known marker genes in conjunction with marker genes identified programmatically with *scran*'s *findMarkers* function[61], and additional exploration with *iSEE*[62]. We identified eight subpopulations that included all major cell types, namely, astrocytes, endothelial cells, microglia, OPC, CPE cells, oligodendrocytes, excitatory neurons, and inhibitory neurons.

DS analysis was run using *edgeR*[22] on pseudobulk (sum of counts), requiring at least 10 cells in at least 2 samples per group for a subpopulation to be considered for differential testing; the CPE subpopulation did not pass this filtering criterion and were excluded from the differential analysis. Genes with FDR < 0.05 and |logFC| > 1 were retained from the output. To distinguish subpopulation-specific and shared signatures, we assembled a matrix of logFCs (calculated for each cell subpopulation) of the union of all differential genes (FDR < 0.05 and |logFC| > 1), and performed consensus clustering of the genes using the *M3C* package[63] (penalty term method), choosing the number of clusters with the highest stability.

To estimate per-cell effect coefficients, we calculated dot products of each cell's normalized log-expression and the group-level logFCs using only the DS genes detected for the corresponding subpopulation.

**Performance summary criteria.** For each of the metrics in Fig. 5, method performances are considered to be "good", "intermediate", "poor", or "NA" (not available). Method assessments were made as follows:

- *TPR/FDR:* For each type of DD category, we consider TPRs and FDR at FDR 5% averaged across two references, five simulation replicates and three clusters (Fig. 2a). Methods are scored according to $\overline{TPR} > 2/3$: good, $> 1/3$: intermediate, otherwise: poor; and $\overline{FDR} < 0.05$: good, $< 0.1$: intermediate, otherwise: poor.
- *null simulation:* We perform a Kolmogorov–Smirnov (KS) test on the uniformity of $p$ values (*ks.test* with CDF $y = $ "*punif*") under the null (Supplementary Fig. 2) for each of two references, three simulation replicates, and three clusters per simulation, resulting in a total of 18 tests per method. KS statistics (largest difference between observed and uniform empirical cumulative distribution functions) are then averaged, and categorized according to $\overline{KS}_{stat.} < 0.1$: good, $< 0.25$: intermediate, otherwise: poor.
- *logFC estimation:* From visual inspection, methods that gave logFC estimates near the diagonal (against the true simulated logFC) were labeled as good; methods with attenuated logFC estimates were listed as intermediate; methods that did not return logFC estimates were given "NA".
- The *complex design* criterion is qualitative. Methods are scored "good" or "poor" depending on whether or not they are capable of accommodating the experimental design of interest, i.e., multiple replicates across two conditions.
- *speed* summarizes the runtimes recorded for increasing numbers of cells and genes (Supplementary Fig. 11). Scores are given according to the three major groups observed (in terms of runtimes) with *scDD* and pseudobulk methods running in the order of seconds; *AD*, *MAST*, and *MM-dream* methods two orders of magnitude longer; and *MM-nbinom* and -*vst* three to four orders of magnitude longer.

Methods were ranked according to the weighted average score across all metrics, with numerical encoding good = 3, intermediate = 2, poor/NA = 0; a weight of 0.5 for error control, logFC estimation, complex design, and speed; and weights of 0.75, 0.125, 0.125, and 0.05 for TPR/FDR on DE, DP, DM, and DB genes, respectively. This weighting of the different DD categories is in accordance with the frequencies of multi-modalities in scRNA-seq data reported by Korthauer et al.[33] (~75% unimodal, ~5% trimodal, and ~25% bimodal, which were split equally between DP and DM).

**Reporting summary.** Further information on research design is available in the Nature Research Reporting Summary linked to this article.

## Data availability

The original droplet scRNA-seq data from Kang et al.[20] is deposited under the Gene Expression Omnibus accession https://www.ncbi.nlm.nih.gov/geo/query/acc.cgi?acc=GSE96583GSE96583. The raw LPS dataset is available from ArrayExpress (accession: E-MTAB-8192) and the Cell Ranger-processed files and metadata are available from https://doi.org/10.6084/m9.figshare.8976473. Both datasets are also available in R through the *muscData* Bioconductor *ExperimentHub* package. Supplementary File 6 is a compressed archive containing R objects of all simulations and results. Supplementary Files 1–6 are available from https://doi.org/10.6084/m9.figshare.8986193.

## Code availability

All analyses were run in R v3.6.2[64], with Bioconductor v3.10[65]. Performance measures were calculated using iCOBRA[66], and results were visualized with *ggplot2*[67], *ComplexHeatmap*[68], and *UpSetR*[69]. All package versions used throughout this study are captured in https://doi.org/10.6084/m9.figshare.8986193 Supplementary File 5. Data preprocessing, simulation, and analysis code are accessible at https://github.com/HelenaLC/muscat-comparison[70], including a browseable *workflowr*[71] website for the LPS dataset analysis (https://doi.org/10.6084/m9.figshare.8986193 Supplementary File 3). All aggregation and DS analysis methods are provided in the *muscat* R package, which is available at https://www.bioconductor.org/packages/muscat through the open-source Bioconductor project.

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

## Acknowledgements

The authors thank members of the Robinson Lab at the University of Zurich for valuable feedback on methodology, benchmarking, and exposition. This work was supported by the Swiss National Science Foundation (grant Nos. 310030_175841 and CRSII5_177208) and the Chan Zuckerberg Initiative DAF, an advised fund of Silicon Valley Community Foundation (grant No. 2018-182828). MDR acknowledges support from the University Research Priority Program Evolution in Action at the University of Zurich.

## Author contributions

H.L.C., C.S., and M.D.R. developed aggregation-based methods; P.L.G. developed MM-based methods. H.L.C. implemented methods, the simulation framework, and the method comparison; C.S. assisted in several technical and conceptual aspects. D.C., L.C., C.R., and D.M. designed mouse LPS experiments; L.C. and C.R. provided mouse cortex tissue sections for snRNA-seq. P.L.G. and H.L.C. performed data processing, analysis, and interpretation; M.D.R. and D.M. assisted in designing analyses and D.M. contributed to interpretation. H.L.C., M.D.R., and P.L.G. drafted the paper, with contributions from all authors. All authors read and approved the final paper.

## Competing interests

L.C., C.R., and D.M. are full-time employees of Roche. H.L.C., C.S., P.L.G., D.C., and M.D.R. declare no competing interests.
