## [Peer Review File · Nature Communications]

In the text below, black text represents the Reviewer's comments and blue text represents our responses.

Reviewer #1:

Thanks for the thorough responses to the points raised. One remaining question I'd still like to see clarified relates to the simulations mentioned below (original comment and response copied below):

"p7. The number of subpopulations in the simulations is typically small ($K = 2, K = 3$) and so there are a ~200 cells / sample / subpopulation in the simulations. Experimental scRNA-seq datasets typically have 5-20+ clusters (subpopulations), so will have fewer cells / sample / cluster (e.g. the LPS dataset has 20 clusters that are combined into 8 subpopulations. Given the improvement in performance observed when going from 50 to 100 cells / sample / subpopulation it could be interesting to explore simulation settings with larger K and/or fewer cells / sample / subpopulation (e.g., 10-20)." [Reviewer's comment first round]

"We thank the reviewer for this comment. It's certainly true that typical scRNA-seq datasets have many more subpopulations than our simulation has used. In terms of understanding the performance across the wide range of possibilities, however, we believe that increasing K while keeping the total number of cells constant will not tell us much more than: fewer cells = lower power. Effectively, more subpopulations would result in an average of the results we already have, weighted by the number of cells per subpopulation. Moreover, the testing for differences of each subpopulation happens independently, so adding more populations would just affect the average performance, and we feel that it is preferred to have the results on a simpler simulation. Furthermore, the synthetic datasets as well as the simulation system are readily available (now with an expanded vignette) for members of the community to conduct their own tests." [Our response first round]

I take the point that simulating more clusters is unlikely to be informative. However, I still think the situation of having fewer cells is of interest. Certainly, one would expect to see reduced power with fewer cells, but can the simulations quantify how much less and how well the various methods are able to cope in scenarios with lower cell number than the current minimum of 50 that is simulated? I imagine that the best methods could still be very useful with smaller sample sizes (e.g., edgeR.sum.counts still has 70-80% TPR while controlling FDR with 50 cells / sample / subpopulation) and simulations with smaller cluster size could shed some light on 'how small is too small' for testing DS on clusters. Perhaps an extra sub panel to accompany Figure 2b (or Supp Fig) with the number of cells / sample / subpopulation set to 20 or so could answer this? While it is indeed true that an appropriately trained reader could run the simulation themselves, I think it would be preferable for the authors to look into this as part of their study and report the results if they reveal any useful insights (and of course ignore them entirely if nothing more than the obvious 'fewer cells = lower power' is demonstrated).

We agree that it would indeed be interesting to know 'how small is too small' and had discussed this at length in the first round of revision. We also feel that it will be difficult to give a fully definitive answer on 'how small is too small', because it depends on many factors. For example, if there is a strong enough effect, even 3-5 cells would probably be sufficient to detect a change. In addition, technologies are changing rapidly, such that data from 20 cells in 2021 could be considerably deeper than 20 cells from datasets of 2019. Nonetheless, we expect that such relative comparisons will be informative.

To address this concern, we have changed the range of the #-of-cells shown in Figure 2b (from 50-100-200-400 to 20-50-100-200-400 cells) to investigate smaller sizes; notably, we changed the visualization of Figure 2b to highlight the change across number of cells for each method (as opposed to a panel for each sample size and all methods on one panel); In particular, this new visualization highlights how FDR control varies drastically by subpopulation size, which we have now commented about in the manuscript.

From a practical perspective, muscat's pbDS() function (for DS analysis on pseudobulks) filters out cluster-sample instances with less than a certain number of cells (default 10), removes samples with a log-library-size 3 MADs below the median log-library-size (presumably because these samples contain only very sparse cells), and removes genes according to edgeR's filterByExpr() function (count of at least 10 in at least some samples, total count greater than 15, gene detected in 70% of samples in the smallest group).

Similarly, mmDS() (for mixed-model-based methods) filters out cluster-sample instances with less than 10 cells, clusters with less than 2 samples per group, and retains genes detected in at least 20 cells. Overall, the stringency of these filtering criteria will determine which genes are subjected to differential testing.

Reviewer #2:

Thank you for the revision and the addition of Supplementary Figure 14. After reading the response to my concerns and the revised manuscript, I must say that my initial concerns about this paper have not changed. I now understand better than the different dots in Figure 2 mean, but still no information is provided about the consistency across the 5 different simulations.

We have now made a plot to show the variability across the 5 simulation runs and have included this as a Supplementary Figure.

Also, I still believe that the performance assessment in Figure 5 is biased towards the taste of the authors. In fact, they indicate that the assessment is visual. If I visually look to, for example, Figure 2, I would have concluded that TPR is not generally good for aggregation methods but probably moderate or even bad, in some cases, especially for DP and DM scenarios. Also, for the definition of error_control based on the uniform p-value distribution under the null hypothesis, I do not see a clear difference between, for example, the plot for edgeR.sum.scalecpm and for MM-vst that would justify that the first gets a “good” and the second an “intermediate” score.

A more formal and measurable definition of these three global assessment categories would have been better here. For example, in terms of FDR, “good” could be when all (or most) simulation scenarios show FDR under the established threshold, “intermediate” when most are at least under 0.1 control, and “bad” when most are above 0.1. For TPR you could establish levels of good (i.e. 0.9), intermediate (i.e. 0.7-0.9) and bad (i.e. < 0.7) and use them in the classification.

We had discussed this comment at length in the original revision and had decided to stick with the simpler visual approach, because methods just above/below a cutoff could in fact still be decently performant methods. After further consideration and given the numerous methods and tests presented, we agree that such a “formal and measurable definition” is a clear advantage; we thank the reviewer for this suggestion. We have now done a more formal performance summary, including a split of TPR/FDR into the 4 types of differential distributions (DE, DP, DM,

DB). The following computations now underlie the new Fig. 5 (see below), and are described under section 'Methods - Performance summary criteria'.

- **TPR:** We score methods according to their TPR at FDR cutoff 5% for each type of differential distribution (averaged across simulation replicates and clusters). To make scores as objective as possible, we evaluate according to $TPR > \frac{2}{3}$ = good, $TPR > \frac{1}{3}$ = intermediate, otherwise poor.
- **FDR:** Similar to the Reviewer #2's recommendation, we score methods according to their FDR at FDR cutoff 5% for each type of differential distribution; according to $FDR < 5\%$ (desired FDR) = good, $FDR < 10\%$ = intermediate, otherwise poor.
- **null simulation:** We perform a Kolmogorov-Smirnov (KS) test against the Uniform distribution of p-values under the null for each reference dataset (2), simulation replicate (3) and cluster (3), resulting in 18 tests per method. For each method, we then average the KS statistics (largest difference between observed and uniform empirical cumulative distribution functions). Methods are scored according to $\text{mean}(\text{KS statistic}) < 0.1$ = good, < 0.25 = intermediate, otherwise poor.
- The assessment of **logFC estimation** and **speed** (runtime) remains as is, and is based on clear differences between the methods in the respective visualizations (Supplementary Figures 4 and 10). Similarly, the **complex design** criterion is unchanged, as it is based on each method's capability to accommodate replicates.
- The final **ranking** of methods is arrived at according to the weighted average of scores with numerical encoding good = 3, intermediate = 2, poor/NA = 0, and the following weighting scheme:
 - error control, logFC estimation, complex design, speed = 0.5
 - $TPR/FDR.DE = 0.75$, $TPR/FDR.DP/DM = 0.125$, $TPR/FDR.DB = 0.05$

where the weighting of the different Dx categories is according to Korthauer et al.'s (DOI: 10.1186/s13059-016-1077-y) reported frequencies of the different number of modalities in scRNA-seq data: ~75% unimodal, ~5% trimodal and ~25% bimodal, which we split equally between DP and DM.

The new Figure 5 is below:

Moreover, there are some results that are puzzling. For example, AD-gid.logcounts shows one of the highest TPR in figure 2A (suggesting high sensitivity), however, it shows concentration of

p.values close to 1 (suggesting poor discovery power) in Supplementary Figure 2a. How is this explained?

Under the null, the AD-gid.logcounts P-value distribution actually has spikes at both 0 and 1 (Supp. Fig. 2a), so it is not a fully clear suggestion of “poor discovery power”. If we look at the P-value distribution of the null simulation as well as the non-null simulation from Fig. 2a (one simulation shown below), we indeed see an increase in the spike at 0, which does suggest that this method has discovery power (of course, as reported, it also suffers from extremely high FDR in some cases).

As simulations shown in Figure 2A are run with 200 cells per subpopulation, I would have expected that the 200 cell panel in Figure 2b would be sort of average representation of values seen in Figure 2b, but for some methods, the FDR values here are worse than any value in Figure 2b, which suggests a modification of additional parameters. Unfortunately, the methods section does not explain in detail how these assessments were performed, and it is hard to understand some of these inconsistencies.

We apologize for the confusion here. The simulations underlying Fig. 2b do not contain an equal mix of the 4 types of differential distributions, thus the panel is not expected to show an average of the results in Fig. 2a. Rather, Fig. 2b is based on “pure” simulations (10% of DE genes), and varying numbers of cells. Thus, results are expected to be most similar between Fig. 2a: DE (left-most panels) and Fig. 2b: 200, since both of these used simulations with ~200 cells per cluster-sample and 10% of DE genes. This was indeed not well described, and we have clarified it both in the main text and figure description.

Finally, my request for performing a comparative analysis of the LPS dataset is poorly addressed. Only upset plots are provided as a Supplementary figure and these are barely discussed. I would have expected a more extensive analysis in just one or two additional methods of the alternative types of approaches where deeper insights into the identification of DS patterns in this dataset by the different methods were provided.

We apologize that our additional analyses seemed to poorly address this concern, but we also would like to reiterate that within the LPS dataset, there is no ground truth, and this puts severe limits on the conclusions that can be made from a more extensive analysis. In particular, it is unclear what can be inferred when some methods detect changes and others do not. Are these false positives or true discoveries?

While it is true there was minimal discussion (in terms of words) about the newly obtained results, we believe that the key findings are summarized well: we observe similar sets of method intersections between both simulated and real data, which also supports the conclusion that the simulation framework faithfully mimics properties of real data and forms a solid basis for the performance evaluations.

We are certainly open to suggestions of analyses that can clearly unravel deeper insights, but we simply think that there are limits to what can be concluded from this dataset, without doing further follow-up experiments (which would be beyond the scope of the current work).

In summary, although I do believe that the muscat package is a great resource for simulation of single-cell data with varying types of DS scenarios, I still feel that presentation of results in this paper does not fully leverage the benchmarking opportunity that the package offers, with many results only discussed superficially. I think this paper could be a much more useful contribution to the field if restricted to the simulation data, a great deal of the performance results now present in the supplementary material would be brought to the main paper together with a more thorough discussion of these results, and a measurable and unbiased score for the final good/intermediate/poor assessment.

We hope that reporting the simulation results at a higher resolution (e.g., TPR split by DE, DP, DM, DB), the formal nature of the good-intermediate-poor classification and the additional discussion have already improved the contribution to the field. In addition, we still believe that we have struck a decent balance between the right “volume” of discussion for the various simulation results presented.

Regarding the comment about restricting to just simulation results, while we cannot draw strong conclusions from the results on the LPS dataset, it still provides a decent case study of the situation, as well as evidence to support the faithfulness of the simulation framework.

Reviewers' Comments:

Reviewer #1:

Remarks to the Author:

I am satisfied with the revisions the authors have made.

Reviewer #2:

Remarks to the Author:

My concerns on previous versions of the manuscript have been sufficiently addressed. I thank the authors for making the effort to create a measurable score to evaluate the performance of the different methods. Also other requested clarifications have been successfully addressed.